# Forecast of International Trade of Lithium Carbonate Products in Importing Countries and Small-Scale Exporting Countries

**Yichi Zhang** , **Zhiliang Dong *** , **Sen Liu, Peixiang Jiang, Cuizhi Zhang and Chao Ding**

School of Management, Hebei GEO University, Shijiazhuang 050031, Hebei, China; ljszyctsl@126.com (Y.Z.); liusen34@126.com (S.L.); jpx815278733@126.com (P.J.); z_19831985185@126.com (C.Z.); hd8516@163.com (C.D.)
* Correspondence: dongzhl@126.com

**Abstract:** As the raw material of lithium-ion batteries, lithium carbonate plays an important role in the development of new energy field. Due to the extremely uneven distribution of lithium resources in the world, the security of supply in countries with less say would be greatly threatened if trade restrictions or other accidents occurred in large-scale exporting countries. It is of great significance to help these countries find new partners based on the existing trade topology. This study uses the link prediction method, based on the perspective of the topological structure of trade networks in various countries and trade rules, and eliminates the influence of large-scale lithium carbonate exporting countries on the lithium carbonate trade of other countries, to find potential lithium carbonate trade links among importing and small-scale exporting countries, and summarizes three trade rules: (1) in potential relationships involving two net importers, a relationship involving either China or the Netherlands is more likely to occur; (2) for all potential relationships, a relationship that actually occurred for more than two years in the period in 2009–2018 is more likely to occur in the future; and (3) potential relationships pairing a net exporter with a net importer are more likely to occur than other country combinations. The results show that over the next five to six years, Denmark and Italy, Netherlands and South Africa, Turkey and USA are most likely to have a lithium carbonate trading relationship, while Slovenia and USA, and Belgium and Thailand are the least likely to trade lithium carbonate. Through this study, we can strengthen the supply security of lithium carbonate resources in international trade, and provide international trade policy recommendations for the governments of importing countries and small-scale exporting countries.

**Keywords:** lithium-ion batteries; international trade; potential partners; link prediction; complex networks



## 1. Introduction

Environmental challenges and energy crises have become unavoidable problems in the development process of most countries. The use and shortage of fossil energy not only exacerbated environmental pollution but also brought conflicts between countries. The energy transition is a key step in solving this dilemma. As a new type of energy, lithium-ion batteries have the characteristics of high energy density, good safety performance, long cycle life, clean and pollution-free, etc., which have attracted great interest and attention. The 2019 Nobel Prize in Chemistry has been awarded to John B. Goodenough, M. Stanley Whittingham, and Akira Yoshino for their contributions to the development of lithium-ion batteries. The rapid development of lithium-ion batteries in recent years has greatly stimulated the demand for lithium. The importance of lithium is increasingly prominent. As the most important intermediate product of lithium ore, lithium carbonate is the basic material for the production of secondary lithium salts and lithium metal. It is at the core of the whole lithium application industry chain and is the most widely and frequently traded product in the international trade of lithium and its minerals. Due to the extremely uneven distribution of lithium resources in the world, lithium carbonate exports are concentrated in a few large-scale exporting countries, which limits the safety of the lithium carbonate

supply, especially in countries with a low endowment of lithium minerals. If there are trade restrictions or other accidents in large-scale exporting countries, how can countries with large import demand for lithium carbonate obtain sufficient supply through other channels? Finding new international trading partners is an effective way to solve this dilemma. Therefore, this research starts with the current situation of industrial research and the development of international trade theory. Based on the research purpose and data availability, the intermediate product lithium carbonate in the industrial chain is selected as the research object. Using link prediction, we predict more reasonable and possible trade relations for countries with large import demand for lithium carbonate, and find more partners to improve the trade security of lithium carbonate products.

In recent years, with the rapid development of the lithium industry, the trade volume of lithium resources has also increased year by year, and the research on lithium resource trade has gradually increased. Some scholars have studied the current status and future development trend of Chinese lithium industry [1]; Zhou (2014) analyzes the global supply and demand prospects of lithium resources and make some reasonable suggestions based on the reserves of lithium resources and the consumption in recent years [2]; Sun (2017) has established a trade-related material flow analysis framework from the perspective of the lithium industry's entire industrial chain to study the global flow of lithium [3]; For international trade research, complex network method is a common method [4], Li (2016) used it in the international trade of lithium carbonate and analyzed the status of Chinese lithium carbonate international trade [5].

In recent years, complex network theory has become a research hotspot in the international academic circles. Due to the intersections and complexity of its disciplines, its fields have expanded from mathematics to computer science and technology [6], biology [7], management [8], physics [9], chemistry [10], economics [11], and other disciplines. For international trade, Du (2017) has used a complex network to study the complex network view of the interrelationship and evolutionary characteristics of international oil trade in 2002–2013 [12]; Yang (2015) has used complex network analysis to study the geographical location of global crude oil flows and its evolution [13]. Moreover, some scholars have combined complex network and link prediction and have studied the link prediction based on the similarity algorithm of link weight allocation for unweighted complex networks [14]. This paper uses the complex network link prediction method to study the international trade of lithium carbonate.

The link prediction problem in network has become a core scientific problem across many subjects, thanks to the academic community's understanding of the importance of network science itself. Link prediction has important applications in network reconfiguration, network evolution model evaluation, recommendation system, etc. [15], and link prediction is widely used in the exploration of aviation networks [16], cooperative networks [17–19], citation networks [20–22], traffic networks [23,24], trading networks [25], criminal networks [26], and bipartite networks [27]. A relatively cutting-edge research direction is to apply link prediction methods to international trade networks, and according to the relationship structure in the trade network, combine network indicators, such as degree, weight, centrality, path length, and common neighbors, with link prediction algorithms to predict the potential trade relations between countries. Guan (2016) conducted the study for the first time. Based on the 2014 international crude oil unweighted non-directional trade network, it takes the number of common trade partners for each country pair as the potential linking motivation to predict the potential international crude oil trade relationship [28]. Secondly, Feng (2017) used link prediction to explore the formation rules of the link relationship in the LPG international trade network [29]; Liu used link prediction to find the next potential link of bauxite international trade [30]. At present, the link prediction of network topology [15] is used to explore the link relationship in the international trade of lithium carbonate, which is a gap in the research on the international trade of lithium carbonate products. At present, no scholar has conducted research on the international trade of lithium carbonate importing countries and small-scale exporting

countries, and the supply security of these countries is always threatened due to the relative lack of lithium resources in these countries and their dependence on large-scale lithium carbonate exporting countries. Based on the above two points of practical significance, this paper uses the link prediction method to predict the potential link of lithium carbonate international trade in importing countries and small-scale exporting countries, and analyzes the lithium carbonate trading network composed of lithium carbonate importing countries and small-scale exporting countries. Features to explore potential trade links formed by actual trade relationships.

In summary, based on the actual situation of national resource reserves, this paper analyzes the trade data of lithium carbonate and the trade role of lithium carbonate trading countries, and deletes the trade data of some big exporters of lithium carbonate. Focus on the study of the trade situation between lithium carbonate importing countries and small-scale exporting countries, excluding the influence of large-scale lithium carbonate exporting countries on its trade. Looking for more trading partners for lithium carbonate importing countries and small-scale exporting countries and looking for its next potential link in the international trade of lithium carbonate.

Based on real trade data from 2009 to 2018, this paper builds an international trade network for lithium carbonate. On the basis of this network, the accuracy algorithm AUC is used for evaluation, and the optimal algorithm is selected from the current four main link prediction algorithms CN, AA, RA, and PA. From the results of the optimal algorithm, find the top 10 potential trade links of each year, we draw a test link and a potential link comparison chart, compare the link prediction result with the actual trade relationship, and evaluate the prediction result. In order to explore the practical significance of potential trade links and make the forecast results more clear and instructive, this paper divides the trade roles of lithium carbonate trading countries. On this basis, the potential trade rules of lithium carbonate are analyzed to predict the countries that are likely to have trade relations in the future.

## 2. Data and Methodology

### 2.1. Data

In recent years, the global electric vehicle industry has been growing rapidly, and the global lithium resource trade flow has increased substantially since 2014. Therefore, this paper chooses 2014 as the midpoint to analyze the global lithium resource intermediate product lithium carbonate trade from 2009 to 2018. The annual lithium carbonate import and export data used in this paper were downloaded from the United Nations Statistics Division [31]. The HS code of lithium carbonate is 283,691, and 10 years of trade data from 2009 to 2018 were selected when downloading. In the process of processing raw data, delete the code does not represent a particular country's trade cooperation data, export and import represent partnerships in the same country, as well as partnership repeat trade relationship. And when exported and imported countries report different volume of transactions, choose a larger trade volume and exclude the smaller trade volume, because the smaller trade volume may be caused by statistical negligence.

### 2.2. Methodology

2.2.1. Construct the International Trade Network of Lithium Carbonate

The purpose of this paper is to predict the future generation of lithium carbonate trade links based on the existing trade relations, in order to find a country pair that is likely to have a trade relationship in the next few years, rather than to explore who will import or export lithium carbonate from where. Thus, we choose to build an undirected network for prediction.

As shown in Figure 1, this paper builds an undirected weighted network for international trade of lithium carbonate based on the method of complex network. The node represents the countries that participate in the international trade of lithium carbonate, the edge represents the actual trade relationship between countries, and the weight of

the edge represents the trade volume of lithium carbonate, the unit is kg. From 2009 to 2018, a network is built each year, a total of 10 complex networks for lithium carbonate international trade.

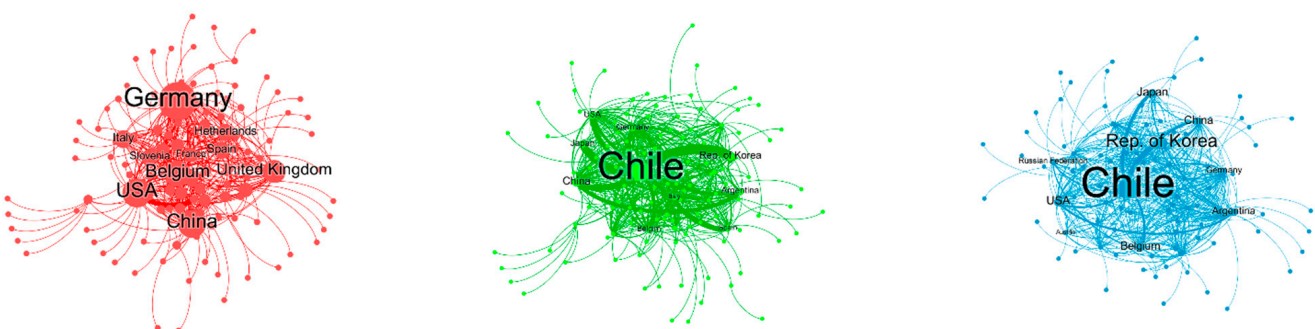

**Figure 1.** Lithium carbonate international trade networks for 2009 (**left**), 2014 (**middle**), 2018 (**right**).

Figure 1 shows the trade relations network of the global lithium carbonate trading countries in 2009, 2014, and 2018. The connection between countries represents the existence of lithium carbonate trade relations between countries. The size of the nodes in the figure represents the number of trading partners. The larger the node, the more trading partners the country corresponds to. According to the size of the nodes, we mark the top 10 trade gathering countries each year. The thickness of the line between nodes represents the amount of trade between two countries. The thicker the line, the more trade between two countries. It can be seen from the figure that the 2009 trade gathering place is located in Germany and the United States, but as time goes by, trade gathering places move closer to Chile. The weight of the Chilean state in the network is significantly higher than other countries. The weight of Germany and the United States has declined, but it is still in the top 10.

It is worth mentioning that in recent years, lithium hydroxide has become an increasingly important part of the lithium supply chain. Similar to lithium carbonate, lithium hydroxide is a compound of lithium, which is located in the smelting and processing link in the middle of the lithium industry chain. International trade in lithium hydroxide is growing as it is the preferred form of lithium for many if not most battery applications. Like lithium carbonate, the outlet of lithium hydroxide is highly concentrated. Chile and Argentina are major producers and exporters of lithium carbonate, and China is the major producer and exporter of lithium hydroxide, accounting for about half of the world's lithium hydroxide exports. The expansion of lithium production in Australia from hard rock mines that is exported to China and converted to lithium hydroxide is a recent important development in the lithium market. Some lithium hydroxide is converted from lithium carbonate. However, increasingly, hard-rock lithium resources are being converted directly into lithium hydroxide, bypassing lithium carbonate altogether (for example, Australian spodumene that is converted into hydroxide in China). It is undeniable that lithium hydroxide occupies an important position in the lithium industry chain, but from an overall analysis, lithium carbonate is the core of the entire lithium application industry chain, and is the most widely used and frequently traded product in the international trade of lithium and its minerals.

### 2.2.2. Link Prediction Model

At present, some scholars have successfully applied the link prediction based on complex network to the analysis and research of international trade [32,33]. This section mainly explains the construction of the link prediction model of lithium carbonate. In order to make readers better to understand the meaning and specific calculation method of each similarity algorithm, this section introduces how to apply four different similarity

definition methods for link prediction through a specific example in Figure 2. Here we consider the simple case of undirected and unweighted networks.

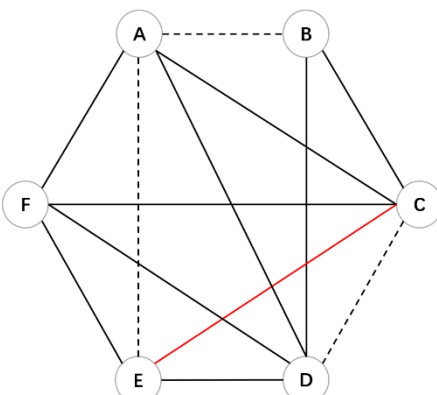

**Figure 2.** Schematic diagram of the link prediction algorithm.

As shown in Figure 2, it can be seen that there are six nodes and 12 edges, among which 10 edges are solid lines, representing the existing links, and three edges are dotted lines, representing the links to be predicted. In order to further test the accuracy of the algorithm, we randomly select 10% of the existing trade links as the test set, here selected {CE} (marked in red). The following four algorithms, CN, AA, RA, and PA, are sequentially selected to perform the similarity calculation to obtain the following results.

The specific steps of this study are as follows:

Step 1: Select the optimal algorithm

At present, there are four mainstream algorithms applied in link prediction research, namely CN, AA, RA, and PA (Table 1), which are based on the similarity index of common neighbors. Their advantages lie in low computational complexity and are suitable for large-scale network applications such as national trade, and these algorithms have proved to be effective in previous studies. In the international trade network of lithium carbonate, if both country B and country C trade with country A, we think that country A is the common neighbor of country B and country C. If two countries are more similar in common neighbor indicators, it is easier for them to establish trade relations. In this paper, the four algorithms are used for calculation, the results of the four algorithms are compared, and the optimal algorithm is selected to explore the network connection characteristics of lithium carbonate international trade. The four algorithms will be briefly introduced in the following with the actual international trade of lithium carbonate.

**Table 1.** Results of the algorithm diagram.

| Rank | Link | CN | Link | AA | Link | RA | Link | PA |
|------|------|----|------|------|------|------|------|----|
| 1 | CD | 4 | CD | 9.17 | CD | 1.42 | CD | 16 |
| 2 | AE | 3 | AE | 4.98 | AE | 0.75 | CE | 12 |
| 3 | AB | 2 | AB | 3.32 | AB | 0.5 | AE | 9 |
| 4 | CE | 1 | CE | 1.66 | CE | 0.25 | AB | 6 |

(1)　CN algorithm

CN similarity can also be called structural equivalence, that is, if two nodes have many common neighbor nodes, then the two nodes are similar. The basic assumption of applying CN algorithm in link prediction is that if two unconnected nodes have more common neighbors, they are more inclined to join edges. In other words, the more common trading partners between two countries involved in lithium carbonate trade, the greater possibility of trade between these two countries. There are two formulas for CN algorithm. One is the unweighted algorithm without considering trade volume, as shown

in Equation (1); the other is the weighted algorithm considering different trade volume, as shown in Equation (2):

$$S_{xy}^{CN} = |\Gamma(x) \cap \Gamma(y)| \tag{1}$$

Returning to the international trade of lithium carbonate, x and y in Equation (1) represent two countries participating in the international trade of lithium carbonate, $\Gamma(x)$ represents a collection of countries with trade relations for country $x$. $\Gamma(x) \cap \Gamma(y)$ represents the number of countries that have direct trade relations with both $x$ and $y$ countries:

$$S_{xy}^{CN} = \sum_{z \in \Gamma(x) \cap \Gamma(y)} \frac{w(x,z)^\alpha + w(z,y)^\alpha}{2} \tag{2}$$

In the international trade of lithium carbonate, $z$ in Equation (2) represents the common trading partner of lithium carbonate trading countries $x$ and $y$ and $w(x,z)$ represents the trade volume between country $x$ and country $z$. $\alpha$ represents the parameter that controls the contribution of trade volume. When $\alpha = 1$, countries with more trade volume play a greater role; when $\alpha = 0$, the algorithm is a case of unweighted, and countries with different trade volumes play the same role. When $\alpha = -1$, countries with smaller trade volume play a greater role.

(2)    AA algorithm

The idea is that the contribution of the common neighbor node with small degree is greater than that of the common neighbor node with large degree. The AA algorithm assigns a weight value to each node according to the degree of the common neighbor node, which is equal to one logarithm of the degree of the node:

$$S_{xy}^{AA} = \sum_{z \in \Gamma(x) \cap \Gamma(y)} \frac{1}{\log k_z} \tag{3}$$

$$S_{xy}^{AA} = \sum_{z \in \Gamma(x) \cap \Gamma(y)} \frac{w(x,z)^\alpha + w(z,y)^\alpha}{\log(1 + S(z))} \tag{4}$$

In Equation (3), $k_z$ is the number of countries with direct links for country $z$. In Equation (4), $S(z) = \sum_{i \in \Gamma(z)} w(z,i)^\alpha$ and $i$ denotes a country that generates trade relationship with the country $z$.

(3)    RA algorithm

Zhou Tao and others proposed the resource allocation algorithm (RA), which is similar to the AA algorithm. In the RA algorithm, considering the two countries $x$ and $y$ without direct trade links, some resources can be transferred from the country $x$ to the country $y$, and in the process, their common neighbors become the medium of transmission. If the impact of trade volume needs to be considered, use Equation (6) to calculate:

$$S_{xy}^{RA} = \sum_{z \in \Gamma(x) \cap \Gamma(y)} \frac{1}{k_z} \tag{5}$$

$$S_{xy}^{RA} = \sum_{z \in \Gamma(x) \cap \Gamma(y)} \frac{w(x,z)^\alpha + w(z,y)^\alpha}{S(z)} \tag{6}$$

(4)    PA algorithm

The more trading partners a country has, the easier it is for the country to establish new trade relationships. As shown in Equation (7). If the impact of trade volume needs to be considered, use Equation (8) to calculate:

$$S_{xy}^{PA} = |\Gamma(x)| \times |\Gamma(y)| \tag{7}$$

$$S_{xy}^{PA} = \sum_{j\in\Gamma(x)} w(x,j)^{\alpha} \times \sum_{q\in\Gamma(x)} w(y,q)^{\alpha} \tag{8}$$

In Equation (8), $j$ represents the country that has a direct trade relationship with country $x$. Similarly, $q$ represents the country that has a direct trade relationship with country $y$.

Step 2: Divide the trade training set and the trade test set.

In order to further test the accuracy of the algorithm, 10% of the existing trade links $E$ were randomly selected as the test set, denoted as $E^T$, and the remaining 90% trade links were selected as the training set [15], denoted as $E^R$:

$$E = E^T + E^R \text{ and } E^T = 10\% * E \tag{9}$$

Step 3: Look for links that do not exist.

Potential trade link relationships usually arise from links that do not exist. We first need to find links that do not exist in order to predict trade. Assuming that there are n countries participating in the international trade of lithium carbonate, Equation (10) can be used to calculate the set $U$ of all trade links among these participating countries. Therefore, the set of non-existent link relations can be calculated by Equation (11):

$$U = \frac{n * (n - 1)}{2} \tag{10}$$

$$E^I = U - E \tag{11}$$

Step 4: Sort the test set and the set without link relation according to the four algorithms.

CN, AA, RA, and PA were used to calculate the test set and the non-existent link relation set, respectively, and the links were sorted according to the score of the algorithm. The higher the score of nonexistent link, the more likely it is to be converted into an existing link in the future, that is, the corresponding country of the link is more likely to have trade relations in the future.

Step 5: Evaluate each algorithm and select the best one.

AUC, ranking score, and precision are three commonly used indicators to measure the accuracy of link prediction algorithms. AUC measures the accuracy of the algorithm as a whole. The ranking score considers the position of the edge in the test set in the final sort. The precision is defined as the percentage of prediction accuracy in the first L prediction edges. The ranking score and the precision only focus on the part of the information [15]. In view of the fact that this paper is based on the whole trade network to explore the causes of the formation of lithium carbonate international trade relations, AUC is chosen as the predictive accuracy evaluation index.

AUC can be understood as the probability that the score value of a randomly selected edge in the test set is higher than the score value of a randomly selected nonexistent edge. That is to say, each time randomly selects an edge from the test set, and then randomly selects one from the non-existing edges. If the score in the test set edge is higher than the score of the edge in the non-existent link relationship set, then one point is added, or 0.5 points if the two sides score equal, independently compared $n$ times. If there are $n'$ times in the test set, the edge score value is greater than the non-existing edge score, and there are $n''$ times that the two scores are equal, then the AUC is defined as [34]:

$$AUC = \frac{n' + 0.5n''}{n} \tag{12}$$

According to Equation (12), obviously, if all scores are randomly generated, AUC $\approx 0.5$. Thus, the degree to which the AUC is greater than 0.5 measures the degree to which the algorithm is more accurate than the randomly selected method. That is, the higher the AUC score, the more accurate its corresponding algorithm.

2.2.3. Analysis Model of the International Lithium Carbonate Trade

Although the link prediction model provides a new analytical perspective on the cause of international trade relations of lithium carbonate, the development status of lithium carbonate industry varies from country to country. In order to explore practical meaning of potential trade links, more accurate and realistic trade relationship forecasts are obtained. This paper constructs an international analytical model of lithium carbonate that links the physical topology features of link prediction with trade characteristics. The steps are as follows:

Step 1: Compare potential links with actual links

The potential trade relations and actual trade relations between 2009–2018 were compared, and the success rate of prediction was calculated to verify the validity of the link prediction model. If most of the potential trade links predicted by the link prediction model are implemented in the next few years, this indicates that the application of this optimal algorithm is effective in practice.

Step 2: Divide the trade role of the predicted potential trading countries of lithium carbonate

In order to further analyze the prediction results, this paper divides the countries or regions involved in the international trade of lithium carbonate into different trade roles. Based on the trade data obtained, the countries that appear in the top 10 of potential trade links are classified as net importers and net exporters. Equation (13) represents the partition rule:

$$D_t = \begin{cases} E_t, E_{Xt} - I_{Mt} > 0 \\ I_t, E_{Xt} - I_{Mt} < 0 \end{cases} \tag{13}$$

In Equation (13), $t$ represents the year, $D_t$ represents the role of a country in the $t$-th year, and $I_{Mt}$ and $E_{Xt}$ represent the import and export volume of a country in the $t$-th year, respectively. If a country has $I_{Mt} > E_{Xt}$ in the $t$-th year, the country is classified as a net importer ($I_t$) in $t$-th year, otherwise the country is classified as a net exporter ($E_t$).

Step 3: Explore trade rules

Combine the results of the first and second steps to explore the rules for the formation of lithium carbonate international trade links. For the next step: to find the most likely links from the potential links of lithium carbonate international trade and provide screening rules.

## 3. Results and Discussion

*3.1. Screen Importing Countries and Small-Scale Exporting Countries from Potential Lithium Carbonate Trade Links*

3.1.1. Select the Optimal Algorithm and Search for Potential Links in the International Trade of Lithium Carbonate

In this section, the accuracy index AUC corresponding to the four algorithms is compared to find the most appropriate algorithm. The higher the AUC score, the higher the accuracy of the corresponding algorithm. Therefore, the algorithm with the highest AUC score was selected for prediction in this study.

Figure 3 shows the AUC scores of the four algorithms. The Y-axis represents the years from 2009 to 2018. The X-axis represents the case where the four algorithms CN, AA, RA, and PA are, respectively, unweighted and weighted, where $\alpha = 0$ indicates that the transaction volume is not considered, and $\alpha = 1$ indicates that the transaction volume is considered. The value of the Z axis represents the AUC score result under the corresponding algorithm in the corresponding year. In order to reduce the random error and make the scoring result more scientific, the AUC score here is the average result of 10 experiments. Each of the 10 experiments randomly divided the annual data into 10% test sets and 90% training sets [35]. By comparing the scores of AUC, it can be concluded that the scores of AUC are all greater than 0.5 and close to 1, indicating that these four algorithms are feasible in the international trade data of lithium carbonate. In addition, the optimal algorithm for each year of the decade is the PA algorithm with $\alpha = 0$. This shows that when using the PA algorithm to predict new trade links, it is equally important

for the formation of new trade relations whether has trading volume or not. That is, in the new trade relationship predicted by the PA algorithm, a country with a large trade volume is as important as a country with a small trade volume.

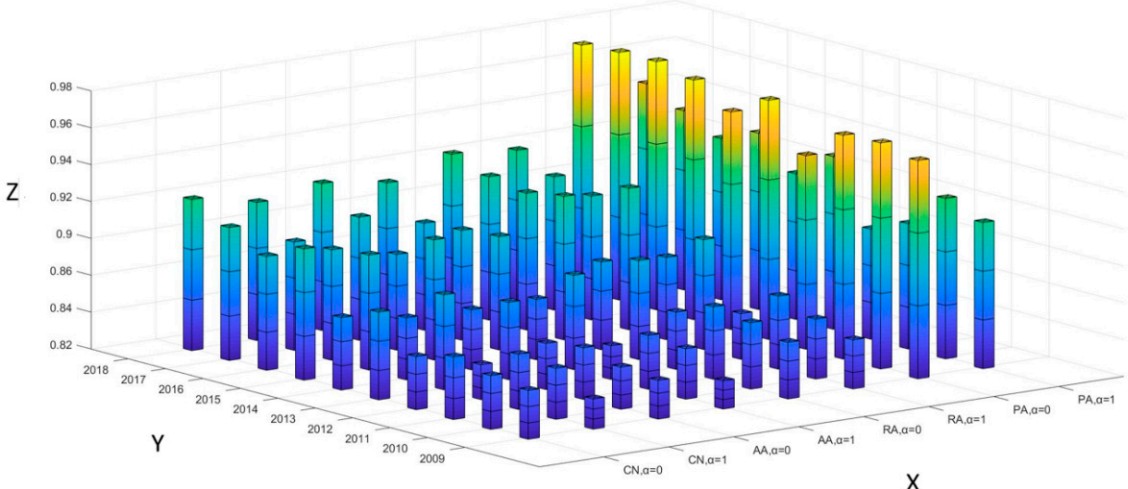

**Figure 3.** Evaluation of four algorithms from 2009 to 2018.

Link prediction is to predict the possibility of creating a link between two nodes that have not yet generated an edge in the network through the information of known network nodes and network structure. It excavates the essence of network formation from the perspective of topological structure, and realizes the prediction of the network through the definition of evaluation index, test set division, unknown edge evaluation, and accurate evaluation of the index. We can judge the accuracy of each algorithm through the AUC indicator. First of all, from the perspective of prediction accuracy, the accuracy of the PA algorithm is higher than other algorithms. The higher the accuracy, the more accurate the prediction. Secondly, from the perspective of algorithm connotation, the PA algorithm is widely used to explore network dynamics. The international trade relationship network is a complex dynamic relationship network, which is not invariable, but changes with time, and this dynamic network relationship is precisely what other algorithms cannot capture. All these support the choice of PA algorithm in this study.

Through the analysis of the accuracy index AUC score results, this paper chooses the PA algorithm without considering the trade volume to carry out the calculation of the discovery of potential trade links. We first sort the test set and the non-existent link relationship by the PA value without considering the trade volume. The definition of the PA algorithm is as follows: the more trading partners, the easier to establish new trade relations, so the countries with the highest PA value are more likely to generate new trade relations than the countries with the latter ranks. For example, in Table 2, the PA value score of the combination of Germany and Belgium is higher than that of Germany and Switzerland, because the degree value of the Belgium node in the lithium carbonate international trade network is greater than that of Switzerland, that is, the number of trading partners of the Belgium is more than that of Switzerland. Therefore, the degree value of Belgium node multiplied by Germany node is greater than that of Switzerland node multiplied by Germany node, so the PA value of Germany and Belgium is higher. Consequently, this study selects the potential pairs of trading countries that are ranked at the top and have not traded before. These potential pairs of trading countries are more likely to have trade relations in the future.

**Table 2.** Top 18 trade links for 2014 PA value.

| Rank | Country or Region A | Country or Region A2 | PA Value | Link's Type |
|------|---------------------|----------------------|----------|-------------|
| 1 | Germany | Belgium | 3408 | T |
| 2 | Chile | Germany | 2769 | T |
| 3 | Germany | Switzerland | 1491 | T |
| 4 | Germany | Austria | 1420 | T |
| 5 | Chile | United Kingdom | 1287 | P |
| 6 | China | Argentina | 1188 | T |
| 7 | China | Denmark | 1080 | P |
| 8 | India | United Kingdom | 1056 | T |
| 9 | Italy | United Kingdom | 1056 | P |
| 10 | Italy | India | 1024 | P |
| 11 | Germany | Rep. of Korea | 994 | T |
| 12 | Chile | Netherlands | 975 | P |
| 13 | Denmark | USA | 880 | P |
| 14 | France | United Kingdom | 825 | T |
| 15 | Belgium | Slovenia | 816 | P |
| 16 | China | Poland | 810 | P |
| 17 | India | Spain | 800 | P |
| 18 | Chile | Denmark | 780 | P |

This study selected the top 10 non-existent links for 2009–2018 each year. Table 2 shows the top 18 trading countries or regions with the highest PA algorithm scores in 2014, including test links and non-existent links. The first column of Table 2 represents the PA value ranking, the second and third columns represent the two countries or regions of the trade link, the fourth column represents the PA value of the trade link country or region, and the fifth column represents the type of the link, T represents the links in the test set, and P represents the potential trade links. In the 18 pairs of trading countries or regions, eight pairs of trading relationships came from the test set. The remaining ten pairs represent links that do not exist (shown in red), which are the ten potential trade links that need further analysis.

3.1.2. The Degree of Export of Lithium Carbonate from Potential Trading Countries is Classified

By arranging the top 10 potential links in the decade of 2009–2018, 42 pairs of countries can be obtained, as shown in Table 3.

The research purpose of this article is for lithium carbonate importing countries and small-scale exporting countries looking for more trade partners, to improve the safety of lithium carbonate trade. Therefore, this paper classifies the export volumes of lithium carbonate from the 42 pairs of countries included in the above potential links, to determine the large-scale lithium carbonate exporting countries which have influence on the prediction of international trade links of lithium carbonate and eliminate them.

According to the downloaded trade data, this paper subtracts the total import trade volume from the total export trade volume of a country in the past ten years. According to the size of the results, countries are ranked in terms of how much lithium carbonate they export. The results are shown in Table 4.

According to the ranking of lithium carbonate export level above, in 26 countries, only Chile and Argentina are large-scale exporters of lithium carbonate. In order to eliminate the influence of large-scale lithium carbonate exporting countries on the trade of lithium carbonate importing countries and small-scale exporting countries, this study chose to delete the trade data of these two countries from the total data, especially studying the trade situation between lithium carbonate importing countries and small-scale lithium carbonate exporting countries. Based on the purpose of this study, this study conducted a second experiment.

**Table 3.** Country pairs included in the potential link of lithium carbonate.

| No. | Country A | Country B | No. | Country A | Country B |
|---|---|---|---|---|---|
| 1 | China | Slovenia | 22 | Italy | United Kingdom |
| 2 | Belgium | Slovenia | 23 | Spain | United Kingdom |
| 3 | Slovenia | United Kingdom | 24 | India | Spain |
| 4 | China | Austria | 25 | Chile | Netherlands |
| 5 | Germany | Australia | 26 | Italy | India |
| 6 | China | Denmark | 27 | Argentina | Italy |
| 7 | Germany | Rep. of Korea | 28 | Chile | Denmark |
| 8 | Chile | China | 29 | China | Italy |
| 9 | Germany | New Zealand | 30 | Chile | United Kingdom |
| 10 | Slovenia | USA | 31 | Canada | Belgium |
| 11 | Denmark | USA | 32 | Poland | USA |
| 12 | Netherlands | India | 33 | Belgium | Singapore |
| 13 | Netherlands | Spain | 34 | Netherlands | Sweden |
| 14 | China | Poland | 35 | Netherlands | South Africa |
| 15 | Germany | Argentina | 36 | Turkey | USA |
| 16 | China | Czech Rep. | 37 | China | Sweden |
| 17 | Canada | Germany | 38 | Argentina | Netherlands |
| 18 | Slovenia | Spain | 39 | Netherlands | Russian Federation |
| 19 | Denmark | United Kingdom | 40 | Rep. of Korea | India |
| 20 | Austria | United Kingdom | 41 | Rep. of Korea | Netherlands |
| 21 | Spain | USA | 42 | France | Rep. of Korea |

**Table 4.** Ranking of export levels of potential lithium carbonate trading countries.

| No. | Country | FA Value | No. | Country | FA Value |
|---|---|---|---|---|---|
| 1 | Chile | 521,928,810 | 14 | United Kingdom | −5,732,160 |
| 2 | Argentina | 151,920,203 | 15 | Austria | −7,278,867 |
| 3 | New Zealand | −63,265 | 16 | Belgium | −7,733,905 |
| 4 | Denmark | −585,253 | 17 | India | −10,014,755 |
| 5 | Singapore | −602,497 | 18 | Canada | −11,945,359 |
| 6 | Australia | −620,094 | 19 | Turkey | −14,606,743 |
| 7 | South Africa | −1,030,514 | 20 | Spain | −15,386,015 |
| 8 | Sweden | −1,515,730 | 21 | Italy | −17,558,812 |
| 9 | Czech Rep. | −1,815,980 | 22 | Germany | −31,681,275 |
| 10 | Netherlands | −2,239,279 | 23 | Russian Federation | −44,253,528 |
| 11 | Poland | −2,307,629 | 24 | China | −72,228,973 |
| 12 | Slovenia | −3,023,313 | 25 | USA | −107,544,303 |
| 13 | France | −3,498,290 | 26 | Rep. of Korea | −151,341,248 |

*3.2. Network Construction and Selection of Optimal Algorithm for Lithium Carbonate Importing Countries and Small-Scale Exporting Countries*

Based on the complex network method, an undirected weighted international trade network of lithium carbonate importing countries and small-scale exporting countries is constructed. Data still reflects ten years, constructing a network every year, to obtain a total of 10 lithium carbonate international trade complex network. Figure 4 shows the global trade relationship network of lithium carbonate importing countries and small-scale exporting countries in 2009, 2014, and 2018 obtained from the second experiment. The size of the node represents the number of trading partners. The thickness of the connection between nodes represents the amount of trade between the two countries. The top 10 trade gathering countries for every year are still marked according to the size of the node. By comparing with the first experiment, it can be found that after removing the two large-scale exporters of Chile and Argentina, the difference in the weight of the top 10 countries in the network is reduced. This shows that the trading network of lithium carbonate is more diverse as a whole, which provides a favorable reference for the next experiment.

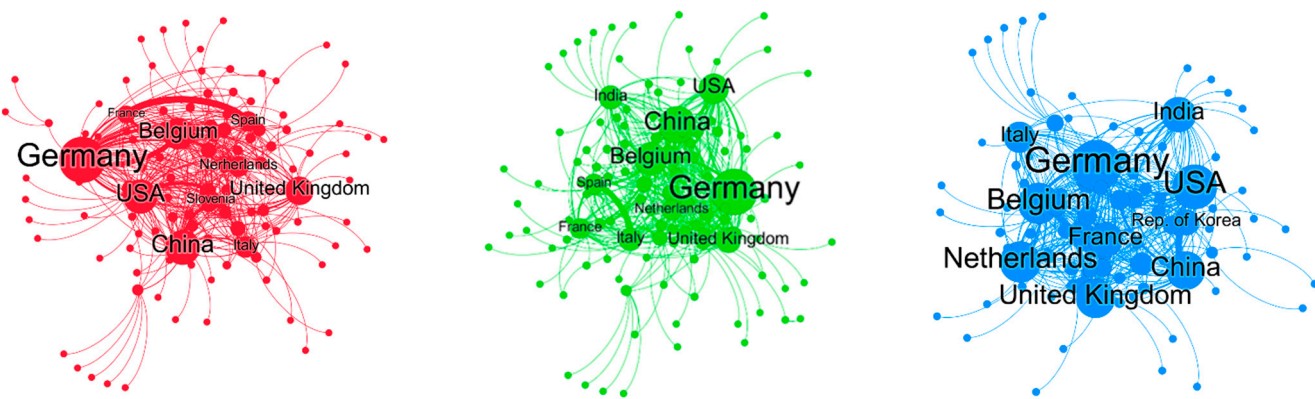

**Figure 4.** International trade network of lithium carbonate importing countries and small-scale exporting countries in 2009 (**left**) 2014 (**middle**) 2018 (**right**).

The trade data of importing countries and small-scale exporting countries are calculated by four algorithms, and the accuracy index AUC analysis is performed on the results of the four algorithms to select the optimal algorithm. As shown in Figure 5, it can be concluded that the PA algorithm still has the highest score without considering trade volume. Therefore, the second experiment still chooses the PA algorithm without considering trade volume to find potential links in lithium carbonate importing countries and small-scale exporting countries in international trade.

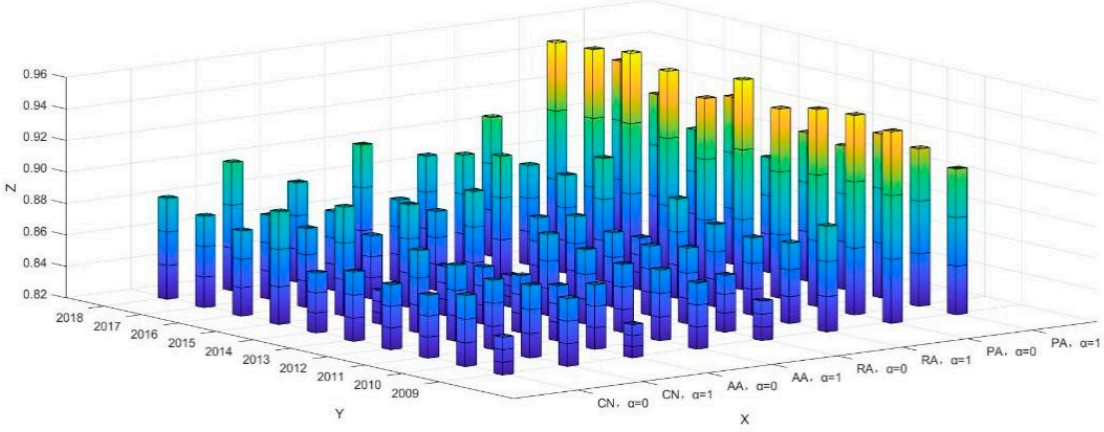

**Figure 5.** Evaluation of four algorithms from 2009 to 2018 (numerical value represents AUC score).

### 3.3. Comparison between Potential and Actual Trade Links and Analysis of Trade Roles of Trading Countries

3.3.1. Select 10 Potential Links

As in the first experiment, in order to explore potential trade rules, the top 10 non-existing links from 2009–2018 were selected and observed whether they actually established trade relations after being predicted to produce trade relations. Table 5 shows the top 22 trading countries or regions with the highest PA algorithm scores in 2014, including test links and non-existent links. By comparing with the first experiment, it can be found that the PA ranking of the tenth potential link dropped from 18th to 22nd, and the PA algorithm's score dropped overall. This is because Chile and Argentina have vast export volumes of lithium carbonate and have established a large number of trades with other countries. After the two countries are excluded from the data, the density of the lithium carbonate trade network will decrease. The weight of the node is reduced, resulting in a lower score for the results of the PA algorithm. This is to explore the trade pattern of

lithium carbonate importing countries and small-scale exporting countries and to obtain the expected result after excluding the international trade situation of large-scale lithium carbonate exporting countries.

**Table 5.** 2014 PA value top 22 trade links (2014 PA value top 22 trade links (importing countries and small-scale exporting countries)).

| Rank | Country or Region A | Country or Region B | PA Value | Link's Type |
|------|---------------------|---------------------|----------|-------------|
| 1 | Germany | India | 2070 | T |
| 2 | Belgium | USA | 1932 | T |
| 3 | China | United Kingdom | 1716 | T |
| 4 | Belgium | India | 1380 | T |
| 5 | China | Netherlands | 1300 | T |
| 6 | Netherlands | Belgium | 1150 | T |
| 7 | China | Denmark | 1040 | P |
| 8 | India | United Kingdom | 990 | T |
| 9 | Italy | United Kingdom | 990 | P |
| 10 | Germany | Poland | 966 | T |
| 11 | Italy | India | 900 | P |
| 12 | Denmark | USA | 840 | P |
| 13 | Germany | Turkey | 828 | T |
| 14 | Belgium | Singapore | 736 | P |
| 15 | China | Poland | 728 | P |
| 16 | France | India | 720 | T |
| 17 | Belgium | Slovenia | 690 | P |
| 18 | India | Spain | 690 | P |
| 19 | China | Finland | 676 | T |
| 20 | Switzerland | United Kingdom | 660 | T |
| 21 | Slovenia | USA | 630 | P |
| 22 | Denmark | Italy | 600 | P |

### 3.3.2. Compare Test Links to Potential Links

Figure 6 shows the results of summarizing the top 10 potential trading countries or regions in each year of 2009–2018. Since there are pairs of countries that repeat in the top 10 in each year, there are 42 pairs of countries after summarizing. The first column of Figure 6 represents the label for the trade country pair, the second and third columns represent the two countries or regions of the trade link, and the 4th to 13th columns represent the years 2009–2018. For each row, the green fill blocks indicates that the country pair has not traded for lithium carbonate for the corresponding year in this column, but since they are in the top 10 ranking of the PA, they are potential trade links. The orange fill blocks indicates that the country pair has produced international trade in lithium carbonate for the year corresponding to this column. The white fill blocks indicate that the country pair has not produced the international trade in lithium carbonate in the corresponding year in this column and is not in the top 10 of the PA value. By observing potential trade links and actual trade links, we can analyze the trend of lithium carbonate international trade in the past decade and provide theoretical support for the next forecast.

The following describes the prediction rules (Figure 6). For each row of corresponding country pairs, if at least one orange fill color block appears after the green fill color block, then the potential trade link indeed has a trade relationship later, and the country pair is the one that predicted successfully. The first country pair in the above picture, for example, China and Slovenia, showed a green-filled color block in 2013, indicating that the country's potential trade relationship was predicted in 2013, followed by an orange-filled color block in 2014. it shows the country pair had actual trade relations in 2014, so the country pair potential trade relationship was predicted successfully. Conversely, If the country pair does not have orange-filled color blocks until 2018 after the first occurrence of green-filled color blocks, it indicates that the potential trade links have not actually traded in the next

few years, which is defined as an invalid prediction. Take the 27th country pair above (Denmark and Italy) as an example. The country pair is an orange-filled color block in 2012 and 2013, indicating that a trade relationship has occurred, and in 2014 it is predicted that a trade relationship will occur, but in other years after 2014, there are white-filled color block, indicating that the country did not have trade relations in other years and was not in the top 10 potential trade links, so the country pair failed to predict.

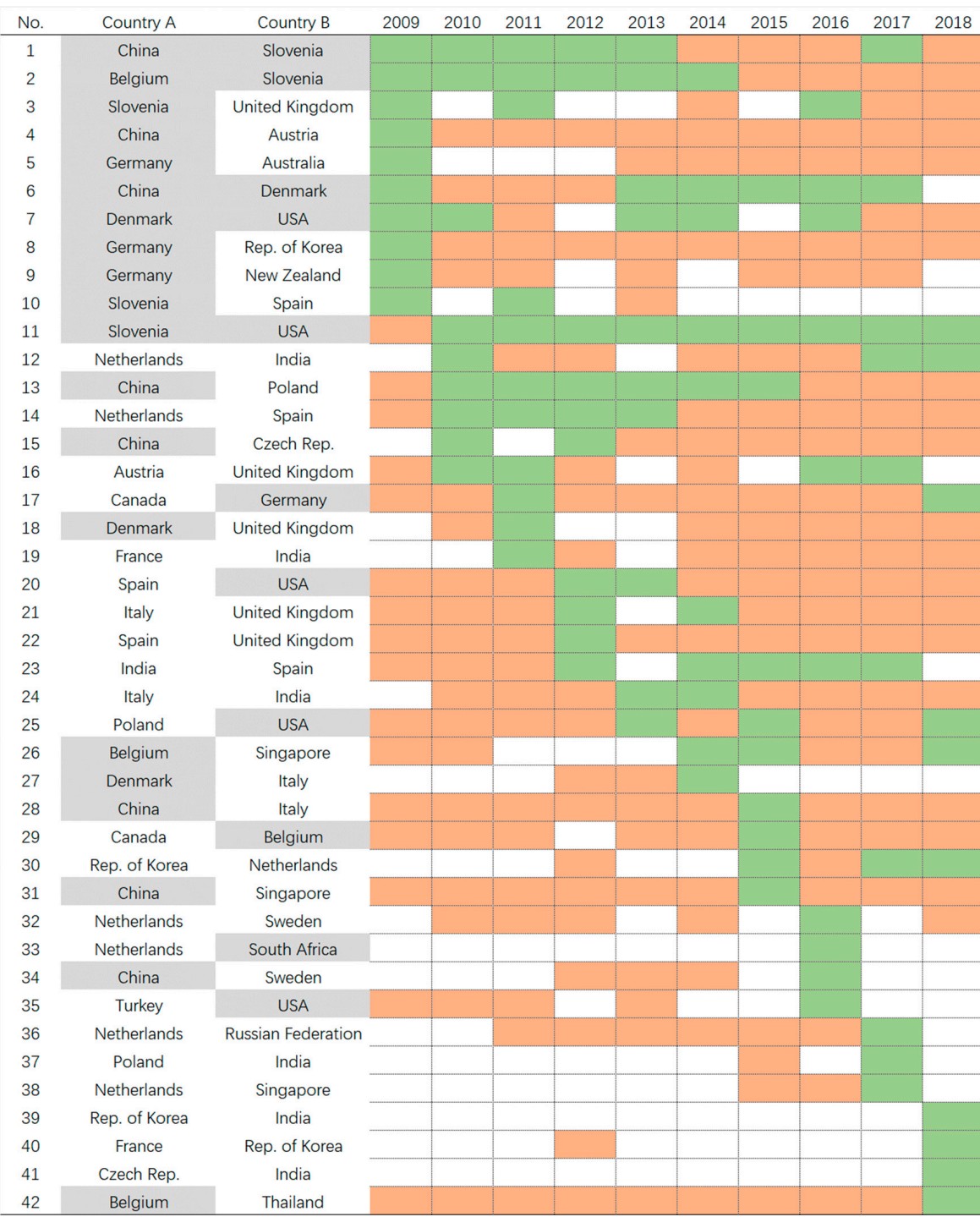

**Figure 6.** Comparison of test links and potential links in 2009–2018 (grey-filled countries are net exporters and unfilled countries are net importers).

The validity of the algorithm can be verified by statistics of the predicted successful country pairs. According to statistics, 29 pairs of countries in the 42 pairs of countries have been predicted to be successful, and the predicted success rate is 29/42 = 69%, which indicates that it is effective to use the PA algorithm to predict the potential links of international trade for importing countries and small-scale exporting countries.

### 3.3.3. Analysis of Trade Roles in Lithium Carbonate Trading Countries

The international trade network of lithium carbonate is complicated. Although PA algorithm has a certain effect in link prediction of lithium carbonate, there are still 11 pairs of trading countries not successfully predicted, which reflects the shortcomings of link prediction based on the physical topology network to some extent. In order to further analyze the prediction results, it is necessary to improve the prediction results by combining the characteristics of international trade of lithium carbonate on the basis of considering its physical topology. Therefore, in this paper, countries or regions participating in the international trade of lithium carbonate are divided into different trade roles, namely, net importers and net exporters of lithium carbonate. By observing trade roles and applying trade rules, to find the links with the greatest trade potential among the 11 pairs of trading countries that have not predicted success.

The trade role of a country can be judged by the quantity of imports and exports. According to the downloaded trade data, we compare the annual import quantity and export quantity of the country or region for ten years to determine the trade role of the country or region in the corresponding year. If a country or region export volume is more than the import volume within one year, the trade role of the country or region in that year is defined as a net exporter. Conversely, if the import volume is more than the export volume, it is defined as a net importer.

Based on the downloaded trade data and the above criteria, this paper summarizes the annual trade roles of all potential trading countries from 2009 to 2018 in Figure 7. The net exporting country is indicated by "E" and filled with blue, and the net importing country is indicated by "I" and filled with gray. If a country or region has not changed its trading role within a decade, it directly defines the trade role of that country or region. For example, Slovenia (no. 1) and Belgium (no. 2) and USA (no. 3) in Figure 7, the three countries have been judged as net exporters in the past ten years, so the final trade roles of these three countries are defined as net exporters. Similarly, countries 17–25 in Figure 7 are defined as net importers. If the trade role of a country or region changes in a decade, this paper considers that the change is caused by its reserves and the amount of exploitation, so those final roles are defined according to the roles of recent years, and are marked in gray. For example, South Africa (no. 5) was defined as a net importer in 2015 and before, but has been defined as a net exporter since 2015. According to recent changes in its role, this study judges it as a net exporter.

### 3.4. Further Classification of Potential Trade Links

By comparing the 2009–2018 test links in Section 3.3.2 with the potential links, it can be found that although 29 of the 42 potential trade links have been come true in the next few years, there are still 13 pairs of links that do not have a real trade relationship even though they are in the top 10. In order to find further rules and obtain further results, this paper classified 42 potential trade links into three categories based on whether they really formed trade relations and the time when potential trade links appeared. Next, these three categories of potential trade links will be further analyzed.

The first category is 29 pairs of countries that have been successfully predicted at least once. As shown in Figure 8, 27 pairs (93%) of the lithium carbonate trading countries had trade relations within five years after their first appearance of the top 10 potential trade links, and the remaining two pairs of trading countries also had trade relations after six years. Thus, we can conclude that if a pair of countries appear in the top 10 potential trade links in a given year, they are likely to generate lithium carbonate trades in the next five to

six years. It also follows that if a pair of countries appear for the first time in the top 10 potential trade links in 2014 or later but have not actually traded lithium carbonate, there is still possibility of a transaction, such as Denmark and Italy, Netherlands and South Africa, China and Sweden, Turkey and USA, Netherlands and Russian Federation, Poland and India, Netherlands and Singapore, Rep. of Korea and India, France and Rep. of Korea, Czech Rep. and India, Belgium and Thailand. These countries are still considered to have a possibility of potential trade links.

| No. | Country | 2009 | 2010 | 2011 | 2012 | 2013 | 2014 | 2015 | 2016 | 2017 | 2018 | Final role |
|---|---|---|---|---|---|---|---|---|---|---|---|---|
| 1 | Slovenia | E | E | E | E | E | E | E | E | E | E | E |
| 2 | Belgium | E | E | E | E | E | E | E | E | E | E | E |
| 3 | USA | E | E | E | E | E | E | E | E | E | E | E |
| 4 | Germany | I | I | E | I | E | I | I | E | E | E | E |
| 5 | South Africa | I | I | I | I | I | I | I | E | E | E | E |
| 6 | Thailand | I | I | I | I | I | I | I | E | I | E | E |
| 7 | Denmark | I | I | I | I | E | E | E | E | E | I | E |
| 8 | China | E | E | E | E | E | E | E | I | I | E | I |
| 9 | Rep. of Korea | I | I | I | I | I | I | I | I | I | E | I |
| 10 | Turkey | I | I | I | I | I | I | I | I | E | I | I |
| 11 | France | E | E | E | I | I | E | E | I | I | I | I |
| 12 | Spain | I | I | I | I | E | E | I | I | I | I | I |
| 13 | Netherlands | I | I | I | E | I | I | I | I | I | I | I |
| 14 | Singapore | E | E | I | I | I | I | I | I | I | I | I |
| 15 | New Zealand | I | E | I | I | I | I | I | I | I | I | I |
| 16 | Italy | E | I | I | I | I | I | I | I | I | I | I |
| 17 | India | I | I | I | I | I | I | I | I | I | I | I |
| 18 | Poland | I | I | I | I | I | I | I | I | I | I | I |
| 19 | Czech Rep. | I | I | I | I | I | I | I | I | I | I | I |
| 20 | Canada | I | I | I | I | I | I | I | I | I | I | I |
| 21 | United Kingdom | I | I | I | I | I | I | I | I | I | I | I |
| 22 | Austria | I | I | I | I | I | I | I | I | I | I | I |
| 23 | Australia | I | I | I | I | I | I | I | I | I | I | I |
| 24 | Sweden | I | I | I | I | I | I | I | I | I | I | I |
| 25 | Russian Federation | I | I | I | I | I | I | I | I | I | I | I |

**Figure 7.** Role division of potential trading countries in 2009–2018.

The second category is pairs of countries that appear in the top 10 potential trade links but never actually have a trade relationship. As Table 6 shows, if a country pair is in the top 10 potential links in a given year, label it with P. If a country has actually traded in a given year, we label it with T. A total of 13 pairs of countries have not formed a trade relationship for ten years, and further analysis is required to understand the reasons why they fail to trade and to find more possibilities for cooperation among potential trading countries.

The third category is based on the second category. These pairs of countries did not appear in the top 10 potential trade links for the first time until 2018. As shown in Table 7, because there is no relevant trade data after 2018, the real trade of the top 10 potential trade links that first appeared in 2018 cannot be verified. Therefore, the four countries listed in Table 7 are not a failure of prediction, but have not been verified by the actual future trade situation. According to the first category, if a pair of countries appear in the top 10 potential trade links in a certain year, they are likely to generate lithium carbonate trade in the next five to six years. Therefore, this paper considers Rep. of Korea and India, France and Rep. of Korea, Czech Rep. and India, Belgium and Thailand. The four pairs of countries are likely to have a trade relationship in the next five to six years.

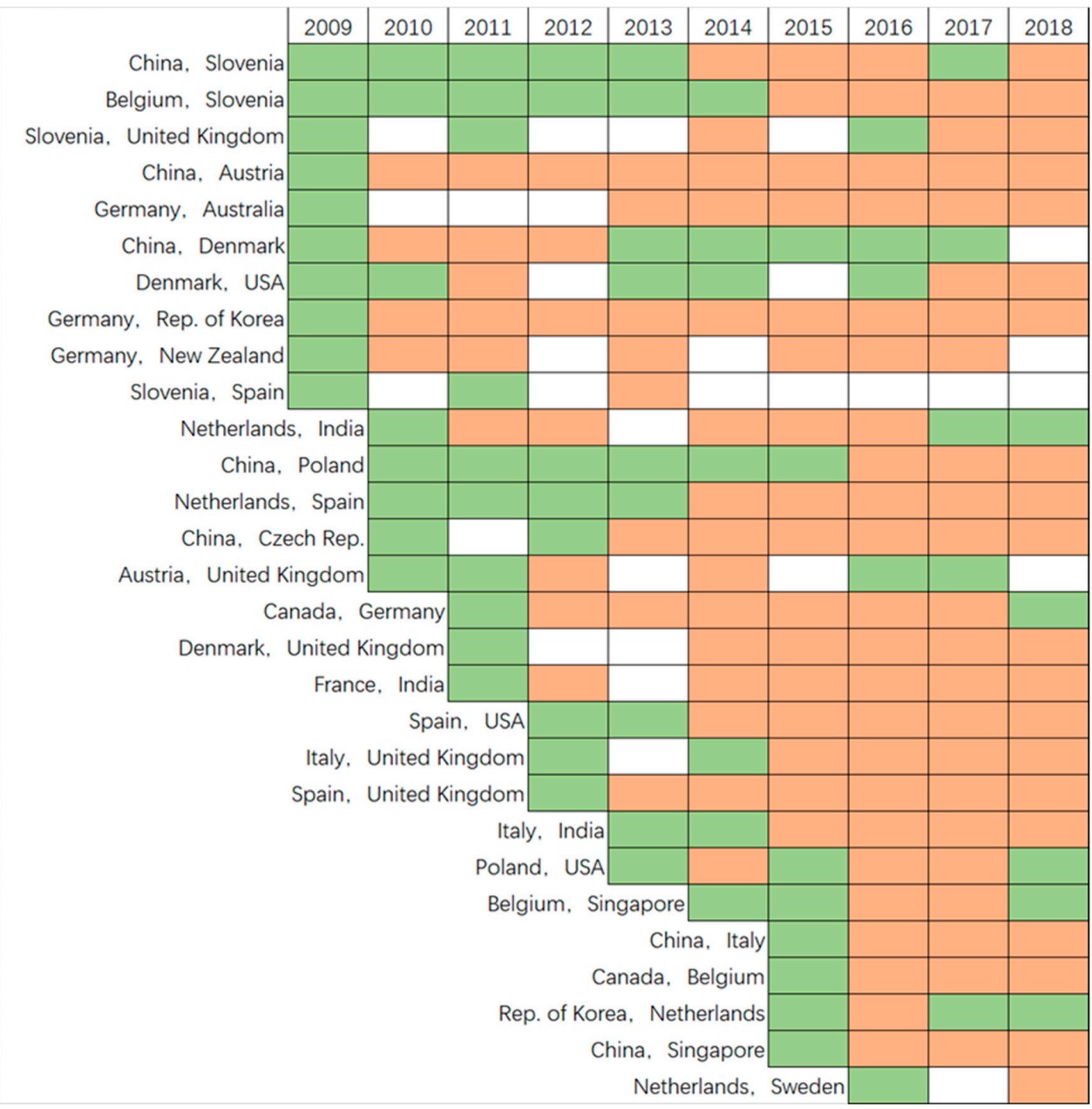

**Figure 8.** Shows the countries that were successfully predicted at least once.

### 3.5. Explore the Linking Rules for Potential Trade

By observing the trade network and PA values, comparing successfully predicted links with unsuccessfully predicted links, combining the division of trade roles and the classification of potential links. The following potential link rules can be summarized. It should be stated that, according to these rules, it is possible to select the most likely trade link from the potential link derived from the topology, but this does not mean that other potential links will not form a real trade relationship. The trade link rules are as follows:

(1) in the potential trade connection, the trade relationship between two net importers is more likely to be established, among which the importers represented by China and the Netherlands are more likely to have new trade relations with other countries.

**Table 6.** Countries that have not been successfully predicted in the top 10 potential trading countries (the countries marked in grey are net exporters).

| Country A | Country B | 2009 | 2010 | 2011 | 2012 | 2013 | 2014 | 2015 | 2016 | 2017 | 2018 |
|---|---|---|---|---|---|---|---|---|---|---|---|
| Slovenia | USA | T | P | P | P | P | P | P | P | P | P |
| India | Spain | T | T | T | P | | P | P | P | P | |
| Denmark | Italy | | | | T | T | P | | | | |
| Netherlands | South Africa | | | | | | | | P | | |
| China | Sweden | | | | T | T | T | | P | | |
| Turkey | USA | T | T | T | | T | | | P | | |
| Netherlands | Russian Federation | | | | T | T | T | T | T | P | |
| Poland | India | | | | | | | T | | P | |
| Netherlands | Singapore | | | | | | | T | T | P | |
| Rep. of Korea | India | | | | | | | | | | P |
| France | Rep. of Korea | | | | T | | | | | | P |
| Czech Rep. | India | | | | | | | | | | P |
| Belgium | Thailand | T | T | T | T | T | T | T | T | T | P |

**Table 7.** Top 10 Potential Trade Links for 2018 (Gray-marked countries are net exporters).

| Year | No. | Pair of Countries | |
|---|---|---|---|
| 2018 | 1 | Rep. of Korea | India |
| 2018 | 2 | France | Rep. of Korea |
| 2018 | 3 | Czech Rep. | India |
| 2018 | 4 | Belgium | Thailand |

Through Figure 6 and in conjunction with Figure 7, it can be found that 22 pairs of countries in the 42 pairs of countries are all net importers. Among them, 14 pairs (64%) of the net importing countries have been successfully predicted to have at least one lithium carbonate transaction. Among the eight pairs of countries without trade relations, three pairs appeared in the top 10 potential links for the first time in 2018, belonging to the third category of potential links mentioned above. This paper argues that the three pairs of countries are likely to have trade relations in the next five to six years. Among the 14 pairs of countries that were successfully predicted to have a trade relationship, China (five pairs) and Netherlands (four pairs) appeared as one of the trading partners. It is therefore concluded that if two trading countries are net importers and one of them is China or the Netherlands, then the possibility of establishing their trade partnership will increase.

China and the Netherlands are major importers of lithium carbonate in the world. Taking China as an example, from the perspective of trade partners, China's trading partners are increasing in the international trade network of lithium carbonate, and its imports are developing in a diversified direction. In terms of import, China's primary lithium carbonate product import market mainly relies on countries are rich in lithium resources, such as Chile and Argentina, but other sources of imports have been quietly transferred from Japan, Italy, Belgium, etc., to Peru, Canada, the United States, and South Korea, etc. This is related to the change of the structure of lithium carbonate imported from China, that is, the increase of battery-grade lithium carbonate and the decrease of industrial-grade lithium carbonate. In terms of the degree of trade amount changing, the Chinese import amount of lithium carbonate was increased from 23.6 million US dollars in 2005 to 66.2 million US dollars in 2014. The import amount of lithium carbonate in China has been growing. Especially since 2011, the import amount and import amount have increased substantially, and both the export volume and export amount are fluctuating forward, those indicate that China is accelerating the adjustment of industrial structure, increasing the development of emerging industries, and increasing the demand for lithium carbonate. In order to meet the requirements of industrial structure adjustment and avoid the impact of external environ-

ment and policies on their own imports, major importing countries began to look for more extensive cooperative relations, which partly explained why China and the Netherlands were easier to establish trade relations with other countries.

(2) Among the potential international trade links of lithium carbonate, if a potential trading country has a trade relationship of more than two years from 2009 to 2018, it is more likely to trade lithium carbonate in the future; if the trade relationship is less than two years, it is less likely.

It can be seen from Figure 6 that among pairs of trading countries (29 pairs) predicted successfully, 27 pairs (93%) had a lithium carbonate trade relationship that is more than two years, which indicates that countries participating in international trade in lithium carbonate are more inclined to trade with trading partners that previously have a trading relationship with them. According to Table 6, among the 13 pairs of countries that were never successfully predicted to have a trade relationship from 2009 to 2018, eight pairs (62%) of lithium carbonate relationships appeared less than two years before the prediction. Therefore, this paper argues those countries that trade lithium carbonate for less than two years are unlikely to have a lithium carbonate trade in the future.

(3) In the potential international trade links of lithium carbonate, the trade relationship between the net importer and the net exporter will soon be realized in the next few years.

Looking at Figure 6, in 42 pairs of countries, 16 pairs of countries consist of a net importer and a net exporter, respectively, and 13 pairs (81%) of these trade links have been successfully predicted at least once. The net exporter is relatively rich in resources, and the net importer has the demand for imports. From the perspective of supply and demand, the trade relationship between the net importer and the net exporter will be easier to realize. Thus, the paper concludes that trade relations are more likely to occur between countries that play different trade roles.

## 4. Conclusions and Recommendations

At present, all countries in the world are actively exploring the development path of the future energy transition, and accelerating the development and utilization of renewable energy has become the universal consensus and concerted action of all countries in the world. Vigorously developing renewable energy can not only reduce dependence on traditional energy sources, but also reduce the emission of harmful gases and substances such as carbon dioxide. As a representative of new energy sources, lithium-ion batteries play an important role in energy saving and emission reduction. Lithium carbonate, as the most important intermediate product in lithium resources, is an indispensable raw material for lithium-ion battery production, and occupies a core position in the whole lithium application industry chain. With the continuous expansion of lithium resource consumption and increasing demand, especially the rapid development of the electric energy vehicle industry [36], as well as the difference in factor abundance of various countries and production technology issues, international trade occupies a large share in meeting the demand for lithium products. However, due to the uncertainty of international trade relations, the supply of lithium carbonate products in some countries is still in a difficult situation. For example, large-scale exporting countries engage in trade monopolies or maliciously increase transaction prices. This would threaten the supply security of the lithium carbonate trading countries with less right to speak. In order to find more trading partners for countries with less say in lithium carbonate trade, this paper uses a new method of complex network theory—link prediction to explore potential international trade links of lithium carbonate—to find the most probable potential trade links through topology-based link prediction and trade rules. According to the results, the following conclusions and recommendations are made:

(1) Importers and small-scale exporters of lithium carbonate should strengthen their trade relations with large-scale exporters of lithium carbonate. From the perspective of supply and demand, lithium carbonate importing countries have a long-term demand for lithium carbonate, and small-scale lithium carbonate exporting countries also need to trade

with large-scale lithium carbonate exporting countries to meet their own demand when the production is insufficient, while large-scale exporter of lithium carbonate have export demand due to their abundant resources and can provide long-term stable supply. From the trade price, combined with the factor endowment theory, lithium carbonate exported by large-scale exporting countries is the product of intensive use of relatively abundant and cheap production factors in this country. Considering the production cost, the price of lithium carbonate exported by large-scale exporting countries should be lower than that of other exporting countries due to their abundant lithium resources. Based on the above two points, it is necessary to strengthen the relationship between importers and small-scale exporters of lithium carbonate and large-scale exporters of lithium carbonate.

(2) According to the conclusions in Section 3.4 above, if a pair of countries appears in the top 10 potential trade links for a certain year, then they are likely to generate lithium carbonate trade in the next five to six years. Therefore, this study chooses to use five years as the time node to sort out the pairs of countries that have not achieved lithium carbonate trade in the top 10 potential trade links in the past five years. As shown in Table 8 (gray represents the net exporting country, T represents the trading of lithium carbonate in the corresponding year, and P represents the predicted lithium carbonate trade in the corresponding year), these 13 pairs of countries are potential trading partners for lithium carbonate. It should be noted that the purpose of this paper is to provide some reference for the government to find new trade partners according to the topological attributes of each country, rather than to determine the exact time when the trade relationship is established. Next, according to the trade rules, look for the most likely lithium carbonate trade links from the 13 pairs of potential trading countries.

**Table 8.** Potential links for lithium carbonate 2014–2018 (Gray-marked countries are net exporters).

| No. | Country A | Country B | 2014 | 2015 | 2016 | 2017 | 2018 |
|-----|-----------|-----------|------|------|------|------|------|
| 1 | Slovenia | USA | P | P | P | P | P |
| 2 | India | Spain | P | P | P | P | |
| 3 | Denmark | Italy | P | | | | |
| 4 | Netherlands | South Africa | | | P | | |
| 5 | China | Sweden | T | | P | | |
| 6 | Turkey | USA | | | P | | |
| 7 | Netherlands | Russian Federation | T | T | T | P | |
| 8 | Poland | India | | T | | P | |
| 9 | Netherlands | Singapore | | T | T | P | |
| 10 | Rep. of Korea | India | | | | | P |
| 11 | France | Rep. of Korea | | | | | P |
| 12 | Czech Rep. | India | | | | | P |
| 13 | Belgium | Thailand | T | T | T | T | P |

Based on the trade rules summarized above, this paper makes the following predictions: Denmark and Italy (No. 3), Netherlands and South Africa (No. 4), and Turkey and USA (No. 6) are most likely to have trade relations in the next five to six years. They comply with Article 3 of the trade rules. A country's trade role consists of a net importing country and a net exporting country. The net exporter is relatively rich in resources, and the net importer has the demand for imports. From the perspective of supply and demand, the trade relationship between the net importer and the net exporter will be easier to realize. Therefore, matching a net exporter with a net importer is more likely than matching two net exporters or two net importers, other factors remaining the same. Secondly, the Netherlands and the Russian Federation (no. 7) are more likely to have a trade relationship in the next five to six years. They comply with both the first and second trade rules, are both net importers, and have actually traded three times in five years. From the perspective of policy factors, the Italian government has encouraged the development of lithium-ion battery industry in recent years, and successfully held the

17th IMLB in Como from June 10th to 14th, 2014. As a country with relatively low levels of an abundance of lithium carbonate, Italy will further expand the import volume of lithium carbonate in order to promote the development of the lithium-ion industry. The EU countries attach great importance to the development of new energy vehicles. The Dutch government announced that it plans to ban the sale of fuel vehicles nationwide in 2025. The demand for lithium is increasing day-by-day, and will further expand the import of lithium resources and increase the capital investment in the lithium-ion battery industry. As an advanced country in the development of the lithium industry in the world, the United States has continuously sought diversification of trading partners in recent years. All those will speed up the realization of potential trade relations. In addition, Slovenia and USA (no. 1), and Belgium and Thailand (no. 13) are the least likely to have a trade relationship. From the analysis of the role of national trade, they are all net exporters of lithium carbonate, which shows that their import demand for lithium carbonate products is relatively small, and there is almost no need to trade lithium carbonate products with each other, so such trade relations are almost impossible to form. The United States is special. It can be seen from Table 4 that the export level of lithium carbonate in the United States is very low, it ranks 25th among 26 countries. However, in the second experiment, the United States has become a net exporter. This is because the processing technology of the lithium industry in the United States is relatively mature. It imports a large amount of upstream products of lithium carbonate in terms of imports, and exports a large amount of downstream lithium products, such as battery-grade lithium carbonate. The large-scale lithium carbonate exporting countries such as Chile and Argentina mostly export lithium carbonate upstream products, and the transaction volume is very large. Therefore, after removing a large amount of data imported from large-scale lithium carbonate exporting countries, the export volume of lithium carbonate in the United States has been highlighted, becoming a net exporter of lithium carbonate.

The above is the potential trade link predicted by the PA algorithm in this paper. If a country does not fall within the scope of the above potential links but wants to find a new lithium carbonate trading partner, the government can first look for potential lithium carbonate trading partners based on the number of national trading partners, and then based on the three trade link rules in the paper, combined with national policies on the import and export of lithium carbonate, to further evaluate potential trade links.

(3) This paper presents the following policy recommendations: for the pairs of countries that have been successfully predicted in 2009–2018 (Figure 8). First of all, these countries should strengthen trade links, consolidate the trade chain of lithium carbonate, maintain good international relations, and minimize the adverse impact of the safety supply and demand of lithium carbonate is caused by political and diplomatic factors. For example, for importing countries, if their main lithium carbonate trading partners have established new trading partners and exported large amounts of lithium carbonate resources due to the impact of new policies or changes in international relations, the importing countries may import less, and the entire lithium industry may be threatened by insufficient supply of intermediate products. Secondly, major trading countries can try to establish trade relations with trading partners of trading countries they are familiar with, increase import and export channels and expand their influence in the international trade market of lithium carbonate, so as to avoid the negative impact caused by single import and export channels and deep dependence on important trading partners. For example, Slovenia and Czech Rep. do not trade in lithium carbonate, but Slovenia and Czech Rep. both have lithium carbonate trade links with China, so these two countries can consider the factor that they have the same trading country to establish lithium carbonate trade relations. Finally, the more lithium carbonate trading partners a country has, the more likely to establish new trade relationships with other countries. Moreover, the method used in this experiment only cares about the existence of trade relations between countries, but the trade volume is not used as a judgment indicator. Therefore, even if the trade volume is

not large, countries should try to establish a lithium carbonate trade relationship with more partners, improve trade security, and increase their voice in the lithium carbonate trade.

Further research should aim at combining to the direction and quantity of the lithium carbonate trade and other multifaceted factors to provide a further perspective, so as to expand the understanding of the international lithium carbonate trade, including more accurate predictions of potential trading partners, and more specific and in-depth evaluation.

**Author Contributions:** Y.Z.: Conceptualization, Methodology, Investigation, Soft-ware, Writing—Original Draft, Writing—Review and Editing; Z.D.: Conceptualization, Visualization, Supervision, Project administration, Funding acquisition; S.L.: Conceptualization, Methodology; P.J.: Data Curation; C.Z.: Resources, Validation; C.D.: Formal analysis. All authors have read and agreed to the published version of the manuscript.

**Funding:** The National Social Science Foundation of China (Grant No. 17BGL202).

**Institutional Review Board Statement:** Not applicable.

**Informed Consent Statement:** Not applicable.

**Data Availability Statement:** The data presented in this study are openly available in [https://comtrade.un.org/].

**Conflicts of Interest:** The authors declare no conflict of interest.

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
