# Peer review of "Forecast of International Trade of Lithium Carbonate Products in Importing Countries and Small-Scale Exporting Countries"

_sustainability, doi:10.3390/su13031251_

Round 1
Reviewer 1 Report
I am not an expert in complex network theory and link prediction models – so I cannot comment on the paper’s methodology. My expertise is in lithium supply chains and markets, and so my comments focus on these aspects of the analysis.
Focusing on international trade in lithium carbonate, this submission sets out to identify existing importing and small-scale exporting countries that (a) do not trade with one another but (b) would benefit from establishing a trade relationship. The analysis uses actual trade data for the period 2009-2018. It identifies 13 pairs of importing and small-scale exporting countries that could benefit from establishing trading relationships, as well as three trading rules or characteristics that are associated with successful intra-sample predictions (that is, predicting actual trading relationships for the period 2014-2018 using results from 2009-2013).
I recommend that the authors be invited to revise and resubmit the paper. The current version is carefully written, but I am afraid it will not be accessible to readers who do are not knowledgeable about network theory and link prediction models.
I recommend revisions in the following areas:
First, throughout the paper, provide more explanation for the general reader on the intuition that explains the basis for the modeling. For example, what is the intuition behind the prediction that Denmark and Italy, the Netherlands and South Africa, and Turkey and the USA are the most likely to establish trade relationships. It is not enough to write (as in line 617) that these country pairs “comply with Article 3 of the trade rules.” What is Article 3 and why does it make sense? The explanation, I think , is simply that among the 13 possible trade partnerships in Table 8, the most likely to actually happen are those involving net importers and net exporters – which makes sense. Matching a net exporter with a net importer is more likely than matching two net exporters or two net importers, other factors remaining the same. In my detailed comments below, I identify other places in the paper that would benefit from greater explanation for the general reader.
Second, in the Abstract and elsewhere, make the trading rules revealed by the analysis in section 3.5 more prominent. In the abstract, for example, state all three rules or situations in which trade relationships are likely to occur: (1) in potential relationships involving two net importers, a relationship involving either China or the Netherlands is more likely to occur, (2) for all potential relationships, a relationship that actually occurred for more than two years in the period in 2009-2018 is more likely to occur in the future, and (3) potential relationships pairing a net exporter with a net importer are more likely to occur than other country combinations.
Third, the authors need to explain their singular focus on lithium carbonate, rather than both lithium carbonate and lithium hydroxide. Focusing only on international trade in lithium carbonate misses an increasingly important part of the lithium supply chain, lithium hydroxide. International trade in lithium hydroxide is growing as it is the preferred form of lithium for many if not most battery applications. Some lithium hydroxide is converted from lithium carbonate. But increasingly, hard-rock lithium resource is being converted directly into lithium hydroxide, bypassing lithium carbonate altogether (for example, Australian spodumene that is converted into hydroxide in China). Chile and Argentina are the major producers and exporters of lithium carbonate, as is noted in this submission. Readers of the submission, however, would not know that China accounts for about half of world exports of lithium hydroxide. I am not suggesting that the authors re-do their analysis to include lithium hydroxide. Rather they need to let readers know that they are aware of developments in lithium markets and acknowledge that their analysis does not include lithium hydroxide.
Fourth, the paper needs careful editing for English. In many places there are incomplete sentences and grammatical mistakes.
ABSTRACT
Rather than simply write that this study help us "understand the link rules of potential trade,” state the specific trade rules that were revealed by the analysis.
INTRODUCTION
Line 29 and elsewhere throughout the paper: The reference to “lithium” batteries is not quite correct. The rechargeable batteries based on lithium are “lithium ion” batteries, the invention of which Goodenough, Whittingham and Yoshino won the Nobel Prize. Lithium batteries, not rechargeable, are a different form of battery.
It is not sufficient to write that the link prediction model is based on the “relationship structure in the trade network” (line 78-79) and “the original common neighbor algorithm” (line 81). Be more specific about what is the relationship structure and a common neighbor algorithm.
DATA AND METHODOLOGY
The text between line 119 and 122, beginning with “On the processing of raw data . . .,” is difficult to understand, and I suspect there is a problem here with the use of English.
Figure 1 and associated discussion: By focusing only on lithium carbonate, the paper misses an important recent development in lithium markets – namely, the expansion of lithium production in Australia from hard rock mines that is exported to China and converted to lithium hydroxide.
Section 2.2.2: A little more explanation is needed for readers not already familiar with the CN, AA, RA and PA algorithms. How do they relate to one another? All seem to be fundamentally based on common neighbors. In the context of lithium carbonate, what is necessary for two countries to be called “neighbors”?
RESULTS AND DISCUSSION
Why is it that the PA algorithm is optimal (section 3)? What are the country characteristics that the PA algorithm captures that the other algorithms do not? Provide enough explanation so that readers who are not experts in link prediction understand the basis for this finding.
Table 2, Table 5, and associated text: Add a statement explaining how readers are supposed to interpret the PA values. Obviously the higher the PA value, the more beneficial a trading relationship would be. But why, intuitively? For example, what are the characteristics that give the Germany-India pair a higher score than, for example, German-Poland?
Author Response
First of all, thank you very much for taking time out of your busy schedule to read and modify my paper. Thank you for your valuable suggestions. You have made an all-round correction on the structure, content, results and research conclusions of my paper, which will play a very important role in the quality of my paper.
We have carefully read your valuable comments and made changes, please refer to the attachment.

Reviewer 2 Report
Dear authors,
This is a very interesting piece of research. Taking into account the relevance of international tradeflows of lithium for the development of power batteries markets, the topic is undoubtly relevant and important. It meets the scope of the journal in terms of the resource provision of sustainable development processes like scaling up electric vehicles.
Authors compare four link prediction algorithms and choose PA to explore existing and possible trade links between the countries. I have no comments to the methodology. It seems fair, the description of all steps is sufficient and the results section is fully match the proposed methodology.
The strength of this study is that the authors don't stop at the determination of the potential trade links and suggest a trading rules/regularities, based on their assessment (Section 3.5). Conclusions are exhaustive and could provide a background for further studies in this area.
Only 2 minor comments:
1.Please, specify software which was used to built fig.1 and fig.4. Looks like MatLAB, but i am not sure. It could be useful for young scientists.
2.Table 3. The table seems too long. I propose to make 6 columns, instead of 3 to show 20 records in each part. №, Country A, Country B ||| №, Country A, Country B.
Author Response

(The authors gave the same response as above.)

Reviewer 3 Report
The paper analyzes lithium carbonate trade links among importing and exporting countries investigated by a link prediction model. The research question is well formulated, methodology applied is adequate, the analysis is precisely carried out, the results are presented clearly. The contributions of the paper are presented. Policy recommendations are included. I did not detect any inappropriate self-citation.
Some minor corrections are needed:
line 119 - HS code is 283691 - code should be explained more clearly
In-text citations include some errors: line 118 - United Nations Statistics Division (https://comtrade.un.org/) - internet link should be replaced to the Reference list.
The formatting of the References should be checked once again before final publishing.
in line 683 - needless star (*) should be removed: [3] Sun X , Han Hao⁎, Zhao F , et al. Tracing global lithium flow: A trade-linked material flow 683 analysis[J]. Resources Conservation and Recycling, 2017, 124:50-61.
line 685: use lowercase instead of capital [4] Maeng S E , Choi H W , Lee J W . COMPLEX NETWORKS AND MINIMAL SPANNING TREES IN INTERNATIONAL TRADE NETWORK International Journal of Modern Physics: Conference Series, 686 2012, 16(01):51-60.
Author Response

(The authors gave the same response as above.)

Round 2
Reviewer 1 Report
I am satisfied with the revised version of the manuscript and recommend that it be published after another round of editing for English.